# upsML: A high-accuracy machine learning classifier for predicting *Plasmodium falciparum var* gene upstream groups

Elcid Aaron Pangilinan[1], Mathieu Quenu[2], Antoine Claessens[2], Thomas D. Otto[1,2*¤a¤b]

**1** School of Infection & Immunity, University of Glasgow, United Kingdom, **2** LPHI, CNRS, INSERM, Université de Montpellier, France

¤a Current Address: Bernhard Nocht Institute for Tropical Medicine, Hamburg, Germany
¤b Current Address: University of Hamburg, Germany
\* thomas.otto@bnitm.de

## Abstract

*Plasmodium falciparum* erythrocyte membrane protein 1 (*Pf*EMP1), encoded by the hypervariable *var* gene family, is central to malaria pathogenesis, influencing both disease severity and immune evasion. Classifying *var* genes into upstream groups (upsA, upsB, upsC, upsE) is important for understanding parasite biology and clinical outcomes, but remains challenging, especially with partial sequences, such as the DBLα tag or RNA-Seq assemblies. We developed upsML, a machine-learning-based classifier trained on 2,530 curated *var* genes, to accurately assign upstream groups based on sequence features from different partial gene regions. We compared seven methods, including support vector machines, random forests, XGBoost, and HMMER models. Several models in upsML achieve accuracies of 83% for DBLα-tag sequences and 92% for full-length *Pf*EMP1 sequences, thereby significantly out-performing existing tools. Additionally, we developed a model to distinguish internal from subtelomeric *var* genes, which we applied to a global collection of *P. falciparum* genomes, revealing a higher frequency of internal *var* genes in Asia. upsML is available at https://github.com/sii-scRNA-Seq/upsML, providing a robust and efficient resource for large-scale *var* gene analysis. It can classify *var* genes from 20 genomes in under one second.

## Introduction

Malaria remains a major global health burden, with *Plasmodium falciparum* responsible for the vast majority of the half-million annual deaths. Transmitted by female *Anopheles* mosquitoes, *P. falciparum* causes disease during its intraerythrocytic life cycle. While early ring-stage parasites are concealed within infected red blood cells (iRBCs), trophozoite-stage parasites export virulence proteins such as *P. falciparum* erythrocyte membrane protein 1 (*Pf*EMP1) to the iRBC surface [1]. *Pf*EMP1

**Data availability statement:** Relevant upsML data for this study are publicly available from the GitHub repository (https://github.com/sii-scRNA-Seq/upsML). All other relevant genome data sources for this study are available within the paper.

**Funding:** TDO was supported by the ExposUM Institute of the University of Montpellier (grants ANR-21-EXES-0005 and Occitanie Region) and the Wellcome Trust: 104111/Z/14/Z & A. EAP was funded by La Caixa Foundation - Health Research Program (HR20-00635). AC was supported by Fondation pour la Recherche Médicale FRM (ANR 18-CE15-0009-01) and Service Programme d'Excellence I-Site, MUSE Pre-ERC, MOTIVATOR (EQU202303016290). The funders had no role in study design, data collection and analysis, decision to publish, or preparation of the manuscript.

**Competing interests:** The authors have declared that no competing interests exist.

enables iRBCs to adhere to endothelial cells via specific binding to host receptors including CD36, EPCR, or ICAM1, allowing parasites to avoid splenic clearance but contributing to microvascular obstruction and inflammation, hallmarks of malaria pathology.

Each parasite expresses only one *Pf*EMP1 variant at a time, due to epigenetically regulated mutually exclusive transcription of *var* genes, the ~60 member gene family that encodes *Pf*EMP1. This controlled expression allows the parasite to switch to a new *var* gene in subsequent cycles, facilitating immune evasion and persistence during chronic, asymptomatic infections [2].

Despite their extreme sequence diversity, *var* genes can be grouped into four major upstream sequence (ups) classes: A, B, C, and E, corresponding to the promoter region upstream of exon 1 [3]. These ups groups correlate with gene location and orientation: group A and E *var* genes are sub-telomeric and transcribed toward the telomere; group B genes are either sub-telomeric or within internal clusters; internal group B genes are also called BC, and group C genes are all internally located. Group E comprises the unusual *var* gene named *var2csa*, which has a nearly conserved sequence. Internal *var* genes are clustered together in six different chromosomal regions that also include other variant surface antigens such as *rif* and *stevor*.

*Pf*EMP1 proteins share a conserved domain architecture: an N-terminal segment followed by combinations of Duffy-binding-like (DBL) and cysteine-rich interdomain region (CIDR) domains, ending with a transmembrane domain and acidic terminal sequence (ATS). Different combinations of DBL and CIDR domains can further be classified into domain cassettes (DCs) [4], with different biological properties and consequences for disease aetiology [5,6]. The DBLα-CIDRα "head" structure is nearly universal and biologically critical, as it determines host receptor binding and thus influences disease severity. For instance, group A and some B variants (DC8) bind to EPCR and are associated with cerebral malaria, whereas group B and C variants bind to CD36 and are linked to milder infections [5–8]. VAR2CSA is associated with pregnancy-associated malaria as it is the ligand binding to Chondroitin Sulfate A in the placenta [9].

The DBLα domain is also widely used as a molecular marker (Fig 1). Due to the presence of conserved motifs, degenerate primers can amplify ~450 bp of the DBLα "tag" region across nearly all *var* genes. This tag exhibits high diversity between parasite clones and provides a scalable tool for population-level studies of *var* diversity and expression, even in the absence of reference genomes [10]. DBLα tag sequencing has proven to be a cost-effective alternative to whole-genome or transcriptome approaches, capable of tracking population structure and *var* expression patterns.

Recently, Tan and colleagues [12] highlighted the use of DBLα tags as a scalable solution in epidemiological surveillance. Thereafter, a novel tool "cUps" was developed to take a short DBLα 'tag' sequence and predict its *var* gene ups group [13]. The tool uses a database of *var* gene tag sequences and links those sequences to ups groups. This was achieved using machine learning (ML) Hidden Markov Models (HMM), which were fitted to the different DBLα subclasses and *var* gene ups groups.

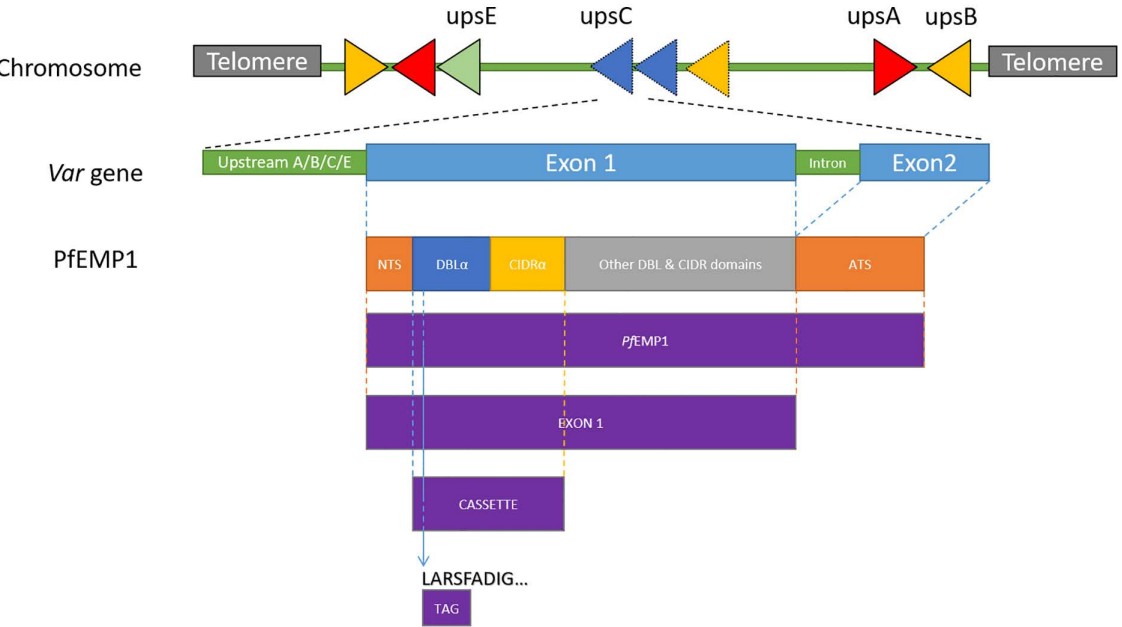

**Fig 1. Structure of *var* gene-encoded *Pf*EMP1 domains. *var* genes can be found in subtelomeric regions (upsA/B/E), and internal *var* gene clusters (upsC/B types).** Most *var* genes comprise an upstream regulatory region and two exons. Exon 1 encodes the extracellular domains, including the N-terminal segment (NTS), and a head structure with a DBLα domain and a cysteine-rich interdomain region (CIDRα). Exon 2 encodes the intracellular acidic terminal segment (ATS). Tag DBLα sequences are conserved fragments of the DBLα domain used in PCR amplification. Domain cassettes are conserved DBLα-CIDRα combinations found across strains. Figure adapted from [11].

The tool has an overall reported accuracy of 74.3% trained on 846 tag sequences [13]. cUps was found to be very accurate at classifying upsA, but lacks accuracy with upsB and upsCs types, consistently.

In light of this, we initially endeavoured to conduct a comparative study of other machine learning techniques to discern the most effective approach for *var* gene annotation. We developed a new tool, "upsML", a suite to run different models for different partial *var* sequences of different lengths and explore alternate algorithms widely used in classification problems:

• Support Vector Machines (SVM): These models aim to find an optimal hyperplane that best separates the data points of the training dataset into different, pre-defined classes or labels. They can take advantage of the kernel tricks, which transform the space of input features into a higher-dimensional space [14].

• Random Forest: an ensemble learning method that combines multiple decision trees. Each tree is trained on a different random subset of the data and features; then, the final classification is based on the majority vote of these trees. It is robust to overfitting and can perform well on imbalanced datasets [15].

• eXtreme Gradient Boosting (XGBoost): a powerful ensemble learning method known for use in imbalanced datasets and sparse training data. It builds decision trees sequentially, where each new tree attempts to correct the errors made by the previous one [16].

We trained these models to link different partial protein sequences encoded by the *var* gene (tag sequences, domain cassette, the protein sequence encoded by the exon 1, and full-length *Pf*EMP1 sequences) to their upstream group on an existing dataset of 2,530 *var* gene sequences.

## Results

### Preparation of *var* gene database

For this study, *var* genes from 66 long-read assemblies were collected (see Methods). As the *var* genes in these assemblies were annotated using different approaches, we re-annotated them as described in the Methods section. On average, 61.7 *var* genes were annotated per assembly, which aligns with the expected ~60 *var* genes per isolate (see S1 Table). An additional dataset comprising 43 isolates from The Gambia was also included [17]. Of the 3,409 *var* genes identified, 2,940 passed our quality control criteria, resulting in an average of 66.8 *var* genes per isolate. These re-annotated and Gambia sequences were translated and yielded a dataset of 4,357 *Pf*EMP1 sequences. We then removed duplicate *Pf*EMP1 sequences using CD-HIT, resulting in a final dataset of 2,530 unique *Pf*EMP1 sequences (see S2 Table).

### Annotation of ups classifications

To classify the upstream sequences of our *var* genes by ups type, we used previously published annotations [4], which included 172 known 600 bp upstream sequences for upsA, upsB, and upsC types. These reference sequences were aligned against our dataset, resulting in confident mapping of all of them to our *var* genes. Sequences annotated as upsE, corresponding to the conserved *var2csa*, were excluded from phylogenetic-based clustering due to their distinct and well-characterised nature.

We then used the annotated sequences from the Rask *et al* [4] as guides to label a phylogenetic tree constructed from all 600 bp upstream sequences in our dataset (Fig 2). The resulting tree showed distinct and well-supported clusters for upsA (red) and upsC (blue). In contrast, upsB-associated sequences (yellow) formed multiple subclades, reflecting higher diversity. Clades containing upsB sequences from the Rask database or in-between such clades were classified as 'upsB'.

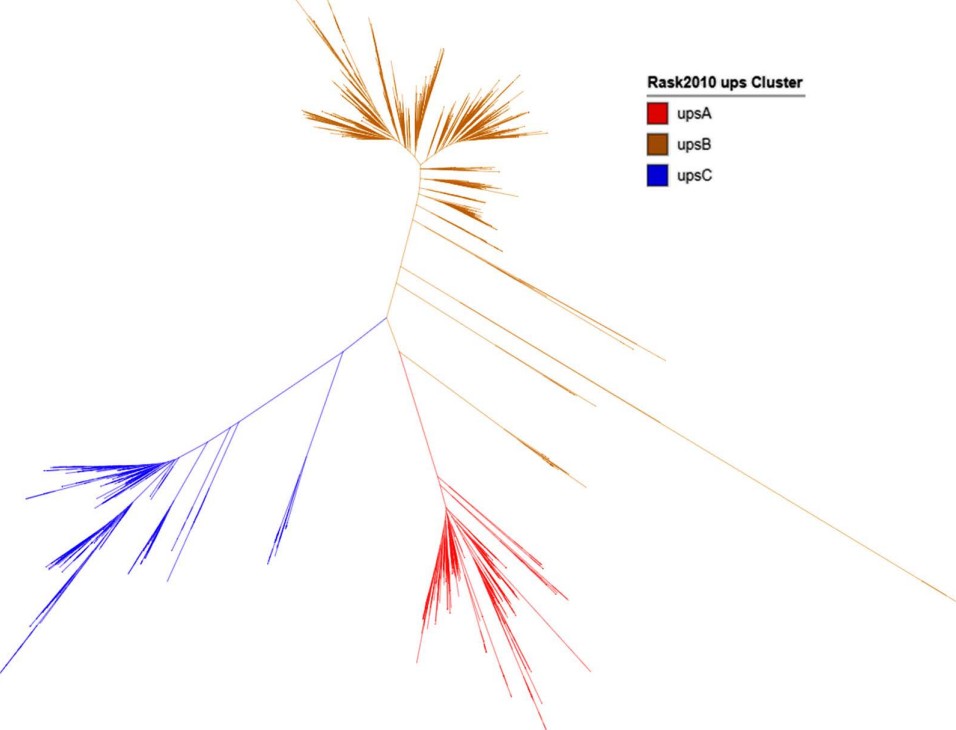

**Fig 2. Phylogenetic tree of upstream 600 bp sequences from *var* genes, including mapped Rask 2010 annotations.** Manual classification of subtrees into upsA (red), upsB (yellow), and upsC (blue) groups was based on consistency with the reference annotations.

After removing duplicates and excluding the 3D7 reference genes, the final dataset included 468 upsA, 1,516 upsB, 507 upsC, plus an additional 39 upsE sequences (total: 2,530). These formed the training and validation dataset used throughout our study.

## Exploration of upsML candidate models

To predict upstream types from *Pf*EMP1 sequences, we tested Support Vector Machines (SVMs), Random Forest (Ran-For), and XGBoost (XGB) across four categories of partial protein sequence types: tag, cassette, the protein sequence encoded by the exon 1, hereafter referred to as 'exon 1', and full-length *Pf*EMP1 sequences. Each sequence set was filtered to remove highly similar sequences, retaining only the longest representative (S2 Table). Furthermore, we excluded the *Pf*EMP1 sequences from the 3D7 reference genome, as these were later used as a ground truth for testing.

An 80/20 split was used for model training and testing, with 80% of the data allocated for training and 20% reserved for evaluation. We explored amino acid k-mers of length 1, 2, and 4 as input features. Best results were obtained with tetrapeptides, we did all further analysis with that model (Table 1 and S3 Table).

The single amino acid features consistently performed worst, with overall accuracies ranging from 0.70 to 0.79. Increasing the k-mer to 2 improved performance to 0.70–0.86. The highest accuracies were obtained using tetrapeptides, with overall performance ranging from 0.75 to 0.91 (S3 Table). The best-performing method varied by sequence type: SVM Poly and RBF yielded the highest accuracy (0.85) for tag sequences, while SVM Linear performed best for cassettes (0.87), see Table 1. For 'exon 1', both SVM Sigmoid and XGBoost achieved the highest accuracy (0.90). For full-length *Pf*EMP1 sequences, SVMs with linear, RBF, and Sigmoid Kernels all achieved an accuracy of 0.92. The differences among models within each sequence type were small (1–2%), which were not statistically significant. Notably, there is a difference in runtime for the models (S4 Table). SVM Linear was consistently over 100 times faster than the others, making it more energy efficient to use. Extending the input sequence length from tags to full-length *Pf*EMP1 sequences increased the accuracy by approximately 2–3% at each step. For instance, the best models achieved 0.85 accuracy for tags, 0.87 for cassettes, 0.90 for 'exon 1', and 0.92 for full-length *Pf*EMP1 sequences.

In summary, no single model consistently outperformed the others. However, using tetrapeptides as input features and longer input sequences consistently yielded the best results. Several models perform similarly; therefore, we recommend linear SVM as it has the lowest runtime.

## Accuracies vary between upstream types

Next, we were interested in understanding how the tools would perform on the four different ups groups. We generated confusion matrices (Table 2) and compared the specificity and sensitivity of each upstream group across all tetrapeptide models for the four sequence types: tag, cassette, 'exon 1', and full-length *Pf*EMP1 sequences (Table 3 and S5 Table).

As expected, upsE classifications were perfect, due to the strain-transcendent nature of the *var2csa* genes. The upsA group was also classified with high accuracy, except in a few cases involving SVM Poly and Random Forest models.

**Table 1. Overall accuracies for classifying different *var* gene sequence regions. Three machine learning models were evaluated: Random Forest (RanFor), XGBoost (XGB), and four SVM kernels: Linear, Polynomial (Poly), Radial Basis Function (RBF), and Sigmoid. Only results using tetrapeptides (k-mer=4) as input features are shown. An 80/20 split was used for training and testing.**

|  | Overall Accuracy | | | | | |
|---|---|---|---|---|---|---|
|  | Linear | Poly | RBF | Sigmoid | RanFor | XGB |
| tag | 0.83 | 0.85 | 0.85 | 0.84 | 0.84 | 0.84 |
| cassette | 0.87 | 0.86 | 0.86 | 0.86 | 0.83 | 0.86 |
| 'exon 1' | 0.89 | 0.88 | 0.89 | 0.90 | 0.88 | 0.90 |
| *Pf*EMP1 | 0.92 | 0.89 | 0.92 | 0.92 | 0.87 | 0.90 |

**Table 2. Confusion matrices of tetrapeptide models across the four ups types and four sequence classes. Green cells indicate correct classifications; red cells indicate misclassifications. "n" denotes the number of sequences used for the assignment.**

| | A n=67 | | B n=261 | | | C n=66 | | | A n=73 | | B n=298 | | | C n=100 | | E n=5 |
|---|---|---|---|---|---|---|---|---|---|---|---|---|---|---|---|---|
| **Tag** | A | B | A | B | C | B | C | **'Exon 1'** | A | B | A | B | C | B | C | E |
| SVM linear | 67 | | 1 | 245 | 15 | 50 | 16 | SVM linear | 73 | | | 285 | 13 | 37 | 63 | 5 |
| SVM poly | 67 | | 1 | 248 | 12 | 46 | 20 | SVM poly | 71 | 2 | | 292 | 6 | 51 | 49 | 5 |
| SVM rbf | 67 | | 1 | 251 | 9 | 49 | 17 | SVM rbf | 73 | | | 285 | 13 | 39 | 61 | 5 |
| SVM sigmoid | 67 | | 1 | 251 | 9 | 53 | 13 | SVM sigmoid | 73 | | | 285 | 13 | 36 | 64 | 5 |
| RanForest | 65 | 2 | 1 | 259 | 1 | 61 | 5 | RanForest | 73 | | | 296 | 2 | 57 | 43 | 5 |
| XGBoost | 67 | | 3 | 241 | 17 | 45 | 21 | XGBoost | 73 | | | 289 | 9 | 40 | 60 | 5 |
| | n=47 | | n=265 | | | n=87 | | | n=98 | | n=296 | | | n=101 | | n=11 |
| **Cassette** | A | B | A | B | C | B | C | **PfEMP1** | A | B | A | B | C | B | C | E |
| SVM linear | 47 | | | 257 | 8 | 44 | 43 | SVM linear | 98 | | | 278 | 18 | 23 | 78 | 11 |
| SVM poly | 47 | | | 258 | 7 | 50 | 37 | SVM poly | 95 | 3 | 1 | 277 | 18 | 33 | 68 | 11 |
| SVM rbf | 47 | | | 253 | 12 | 42 | 45 | SVM rbf | 98 | | 1 | 280 | 15 | 24 | 77 | 11 |
| SVM sigmoid | 47 | | | 254 | 11 | 45 | 42 | SVM sigmoid | 98 | | | 281 | 15 | 24 | 77 | 11 |
| RanForest | 47 | | | 260 | 5 | 62 | 25 | RanForest | 97 | 1 | 3 | 274 | 19 | 45 | 56 | 11 |
| XGBoost | 47 | | | 255 | 10 | 44 | 43 | XGBoost | 98 | | | 279 | 17 | 32 | 69 | 11 |

**Table 3. Specificity and sensitivity of tetrapeptide models. Sensitivity refers to the proportion of true positives; specificity refers to the proportion of true negatives. Top-performing models are shown. Full results are available in S5 Table.**

| | | tag | | | cassette | 'exon 1' | | | PfEMP1 | | |
|---|---|---|---|---|---|---|---|---|---|---|---|
| | | Linear | Poly | RBF | Linear | Linear | Sigmoid | XGBoost | Linear | RBF | Sigmoid |
| A | Sensitivity | 1.00 | 1.00 | 1.00 | 1.00 | 1.00 | 1.00 | 1.00 | 1.00 | 1.00 | 1.00 |
| | Specificity | 1.00 | 1.00 | 1.00 | 1.00 | 1.00 | 1.00 | 1.00 | 1.00 | 1.00 | 1.00 |
| B | Sensitivity | 0.94 | 0.95 | 0.96 | 0.97 | 0.96 | 0.96 | 0.97 | 0.94 | 0.95 | 0.95 |
| | Specificity | 0.62 | 0.65 | 0.63 | 0.67 | 0.79 | 0.80 | 0.78 | 0.89 | 0.89 | 0.89 |
| C | Sensitivity | 0.24 | 0.30 | 0.26 | 0.49 | 0.63 | 0.64 | 0.60 | 0.77 | 0.76 | 0.76 |
| | Specificity | 0.95 | 0.96 | 0.97 | 0.97 | 0.97 | 0.97 | 0.98 | 0.96 | 0.96 | 0.96 |
| E | Sensitivity | | | | | 1.00 | 1.00 | 1.00 | 1.00 | 1.00 | 1.00 |
| | Specificity | | | | | 1.00 | 1.00 | 1.00 | 1.00 | 1.00 | 1.00 |

However, 0.39% of upsB sequences were misclassified as upsA when predicting from tag sequences, and 0.35% were misclassified when using full-length PfEMP1 sequences. Most errors occurred in the classification of upsC. Approximately 25–75% of upsC sequences were misclassified as upsB, depending on the model and sequence type. This may be due to the higher prevalence of upsB in *P. falciparum* genomes, which made them over-represented in the training datasets. Random Forest and XGBoost are often reported to perform better on unbalanced datasets. However, in this case, we did not observe any substantial advantage over the simpler SVM models.

These misclassifications were further examined through per-group specificity and sensitivity analysis (Table 3). Across all models, upsB predictions tended to yield more false positives (i.e., lower specificity), while upsC predictions produced more false negatives (i.e., lower sensitivity). The low specificity of upsB was due to a substantial proportion of upsC sequences being misclassified as upsB. Conversely, upsC predictions were highly specific (specificities above 0.92),

indicating that when a sequence was predicted as upsC, it was almost always correct. However, sensitivity was lower, meaning many true upsC sequences were not detected and instead classified as upsB.

Notably, the prediction accuracy using full-length *Pf*EMP1 sequences reached 92%, compared to 90% using 'exon 1' only (Table 2). When including exon 2, upsC classification improved, while upsB classification slightly decreased: 78 upsC sequences were correctly classified from full-length *Pf*EMP1 sequences compared to 64 from 'exon 1'. This suggests a possible linkage between ups type and exon 2 in internal upsC *var* gene clusters.

Overall, longer sequence inputs yielded better classification performance. SVM Poly and XGBoost offered the best trade-off between accuracy and sensitivity. However, for tag sequences, misclassification between upsB and upsC still occurred in approximately 15% of cases.

### Comparison of upsML with cUps

Currently, no tool predicts the upstream type from a full-length *Pf*EMP1 sequences, or other partial sequences, apart from tag sequences. The most recent method, cUps [13], uses tag sequences as input and applies HMM-based classification, trained on 846 tags with a leave-one-out cross-validation (LOOCV) accuracy of 0.74 (Table 4). To enable a direct comparison, we retrained our classifiers using the same 846-tag dataset and implemented LOOCV. Using tetrapeptide features, our SVM models (RBF and Sigmoid kernels) achieved an accuracy of 0.84, which is 10 percentage points higher than cUps. This performance is comparable to that obtained with our larger training set (~2500 sequences), prompting us to assess the uniqueness of the cUps dataset. In applying CD-HIT with parameters similar to those used on our unique tag set, we found that only 700 of the 846 sequences were unique, indicating 17% redundancy. This helps explain the similarity in performance despite the smaller dataset.

In a complementary analysis, we evaluated cUps using our dataset, with an 80/20 training/testing split (as in Tables 1–3). For ups types A, B, and C, cUps achieved sensitivities of 1.0, 0.5, and 0.42, respectively. In comparison, upsML achieved sensitivities of 1.0, 0.9, and 0.3. The specificities were 0.91, 0.83, and 0.67 for cUps, versus 0.99, 0.65, and 0.96 for upsML. Although individual metrics appear comparable, the overall accuracy of cUps was 0.59, significantly lower than upsML's 0.83. This discrepancy could be due to cUps performing better on the underrepresented upsC class, which contributes less to the overall accuracy due to class imbalance.

### Validation on reference 3D7 genome

As further validation, we tested upsML on the well-annotated *P. falciparum* 3D7 reference genome (Fig 3). This serves as a strong baseline due to the high confidence in the annotated *ups* group labels. As expected, all models accurately classified *upsA* and *upsE* sequences. Most misclassifications occurred between *upsB* and *upsC*, consistent with previous results.

**Table 4. Comparison of cUps with upsML on "cUps dataset". Using the leave-one-out cross-validation accuracies with 846 Tag Sequences of cUps and the upsML methods. Reported is the overall accuracy.**

| | cUps Dataset (846 Tag sequences) | | | | | |
|---|---|---|---|---|---|---|
| | cUps (HMM) | | | | | |
| | 0.74 | | | | | |
| | Linear | Poly | RBF | Sigmoid | RanFor | XGBoost |
| **Amino Acid** | 0.74 | 0.77 | 0.78 | 0.74 | 0.80 | 0.80 |
| **Dipeptide** | 0.79 | 0.83 | 0.84 | 0.78 | 0.83 | 0.83 |
| **Tetrapeptide** | 0.83 | 0.83 | 0.84 | 0.84 | 0.82 | 0.81 |

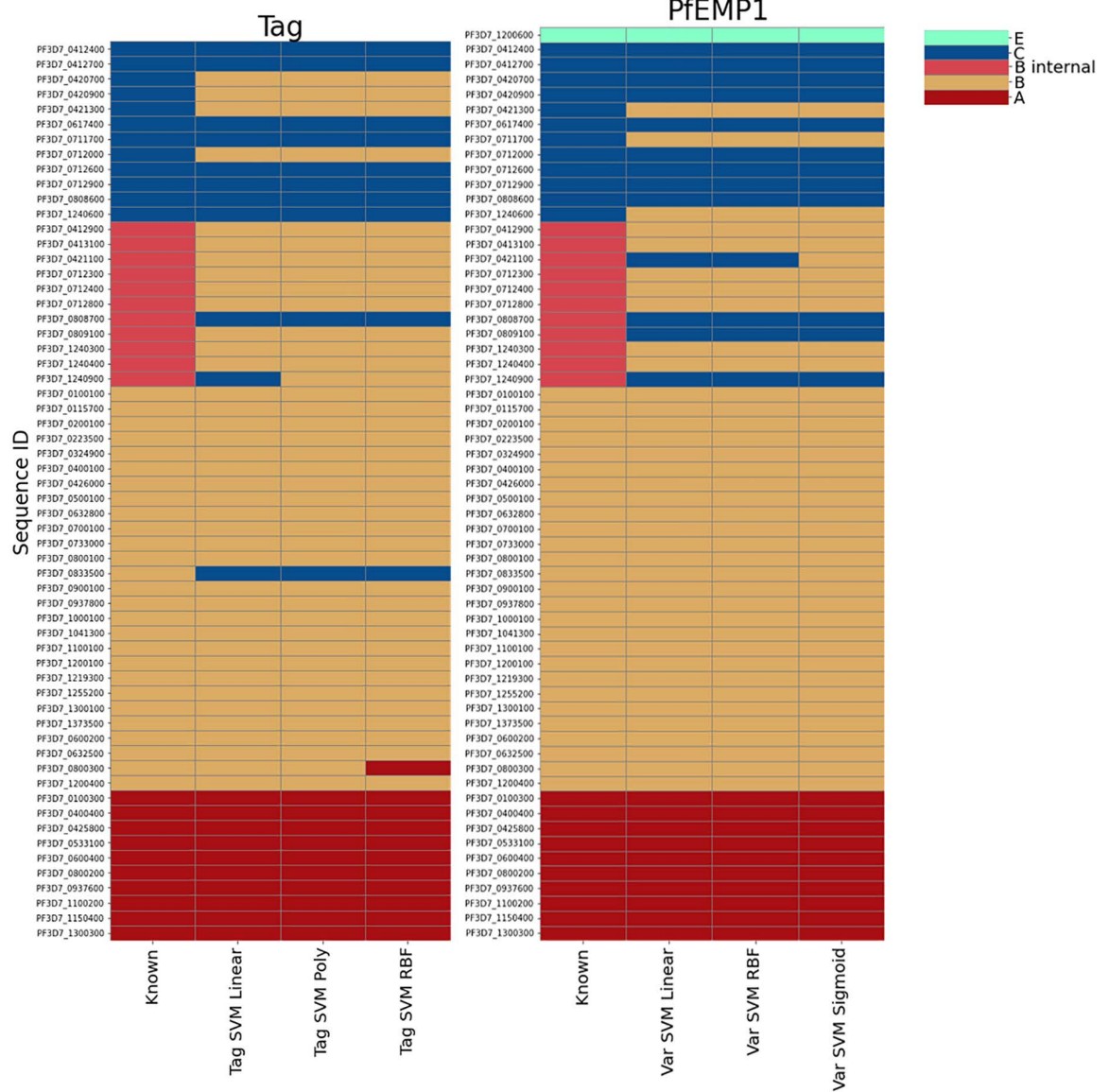

**Fig 3. Classification of 3D7 *var* genes for tag sequences and full *Pf*EMP1 sequences.** "Known" indicates the ground truth. The models used here are the highest accuracy models for each category, except for the Tag SVM linear, as this was included as the fastest model.

For example, using the SVM linear model on tag sequences, three *upsB* sequences were misclassified as *upsC*, and four *upsC* sequences as *upsB*. Interestingly, several *upsC* genes, *Pf3D7_0412700*, *Pf3D7_0420900*, and *Pf3D7_0712000*, were classified differently depending on whether the tag or the full sequence was used. Only one sequence, *Pf3D7_0461300*, was misclassified with both sequence types. This suggests that combining tag- and full-sequence models may improve overall classification performance.

From a biological perspective, such discrepancies may reflect recent recombination events between *var* gene groups in 3D7 that differ from those in the training set. Additionally, we observed that *upsB* classifications were nearly perfect, with errors limited to the so-called BC-type genes – *internal upsB* variants. In these cases, upsML misclassified three internal

*upsB* sequences as *upsC*. These results led us to hypothesise that it may be possible to develop a classifier to distinguish *var* genes based on chromosomal location, specifically, internal *var* genes versus subtelomeric ones.

### A novel "positional" classifier for genome location

To further investigate classification errors (Table 2), we found that 90% of misclassifications occurred between *upsB* and *upsC* genes located within internal *var* gene clusters. This supports our hypothesis from the 3D7 analysis: that recombination events are more frequent within internal clusters [11], and thus these sequences may form a biologically meaningful composite class (Fig 3).

Based on this rationale, we re-annotated 1,980 of the 2,530 genes into four categories: *upsA*, *upsE*, *upsB_subtelomeric*, and *upsBC_internal*, as detailed in the Methods. We then ran all tetrapeptide-based models on this subset (Table 5). Overall, the classification performance for tags remained limited, three percentage points lower than with the first model (Table 1). This slightly worse result could be attributed to the reduced training set size (~75%).

The best overall tag accuracy reached 0.78, with sensitivity for distinguishing *B_subtelomeric* from *BC_internal* ranging from 0.60 to 0.83 (S6 and S7 Tables). Using 'exon 1' sequences improved sensitivity to 0.67–0.85 and specificity to 0.80–0.91. Notably, full-length *Pf*EMP1 sequences yielded results up to 10 percentage points higher, but these are less relevant given that long-read assemblies are still required for complete *var* reconstruction and would indicate location.

Importantly, while accuracy for tag sequences is lower compared to the original classifier (Table 1), classification of *upsA* and *upsE* remains near-perfect. We then compared performance across sequence types (Table 5, S6 and S7 Tables). For tag sequences, although the overall accuracy dropped, classification between *B_subtelomeric* and *BC_internal* was more balanced than in the previous model. For instance, the polynomial SVM yielded sensitivity and specificity between 0.58 and 0.88. Most predictions were correct, contrasting with earlier results where misclassification between *upsB* and *upsC* was frequent.

As expected, performance improved with 'exon 1' sequences, where over 75% of sequences per class were correctly classified. While full-length *Pf*EMP1 sequences yielded the highest performance (>85% accuracy), 'exon 1' might be the most practical input type. Here, overall accuracy reached 0.84, with *B_subtelomeric* sequences classified at >83% sensitivity and 0.90 specificity using the linear SVM model.

In summary, although overall accuracy is comparable to the previous model, classification is more balanced across classes. Moreover, this analysis introduces a novel classifier for predicting the chromosomal position of *var* genes.

### Regional tetrapeptide contributions reveal internal recombination signals

When testing upsML on the 3D7 reference genome, we observed that certain sequences were classified differently depending on whether the full-length *Pf*EMP1 sequence or only the tag sequence was used (Fig 3). For example, *Pf3D7_0420900* has an upstream region classified as upsC, and the full-length model correctly assigned it as such.

**Table 5. Overall accuracy of "positional" classifier to predict internal and subtelomeric *var* gene location. The same ML models were used with a reannotated subset of *var* genes classified as upsA, upsE, upsB_subtelomeric, and upsBC_internal.**

| | Overall Accuracy | | | | | |
|---|---|---|---|---|---|---|
| | TRAIN/TEST 80/20 | | | | | |
| | Linear | Poly | RBF | Sigmoid | RanFor | XGB |
| **tag** | 0.76 | 0.78 | 0.78 | 0.76 | 0.78 | 0.74 |
| **cassette** | 0.81 | 0.80 | 0.82 | 0.81 | 0.81 | 0.80 |
| **'exon 1'** | 0.84 | 0.83 | 0.84 | 0.83 | 0.82 | 0.84 |
| ***Pf*EMP1** | 0.92 | 0.92 | 0.92 | 0.92 | 0.89 | 0.92 |

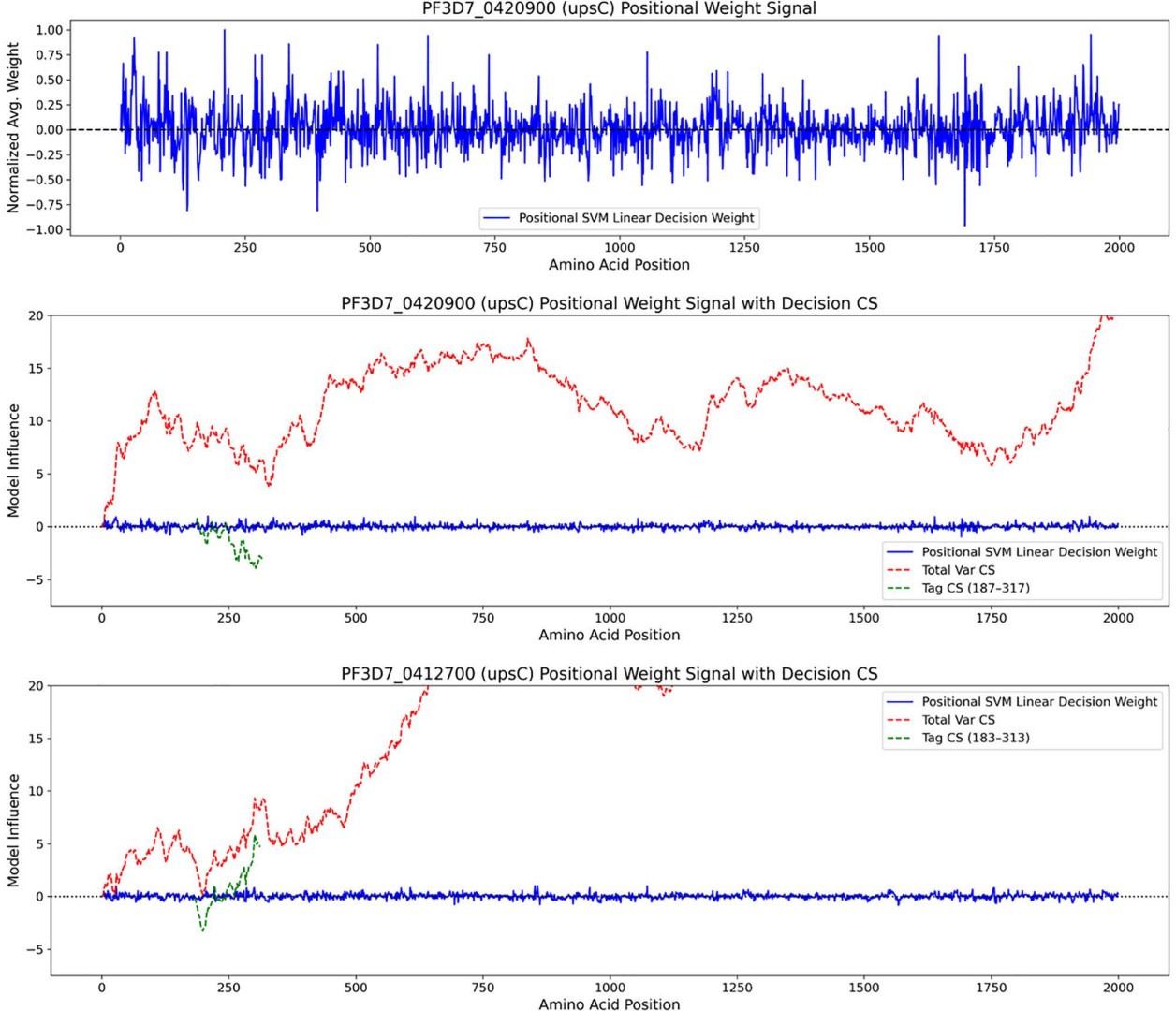

However, when only the tag sequence was used, the tag model classified it as upsB. A notable advantage of using SVM models is the interpretability of their decision function: the contribution weights of individual tetrapeptides can be extracted. This allows us to explore which regions of a gene – and which specific tetrapeptides – are most informative for classification into ups groups.

In Fig 4, we visualise the tetrapeptide decision weights across the sequence of *Pf3D7_0420900*. In the top panel, the decision weights fluctuate along the length of the gene, suggesting no single dominant signal, indicative of a complex or mosaic sequence. To interpret this more systematically, we calculated the cumulative sum of the positional weights

**Fig 4. Regional contributions of tetrapeptides to *ups* classification.** Top panel: Decision weights for each tetrapeptide along the sequence of *Pf3D7_0420900*, generated by the SVM classifier. Middle and bottom panels: cumulative sum (CS) of the positional weights decision scores for two genes, *Pf3D7_0420900* (middle) and *Pf3D7_0412700* (bottom). Red lines represent cumulative scores from the full-length gene; green lines represent scores from the tag sequence. Positive values indicate stronger support for upsC, while negative values indicate stronger support for upsB. Localised shifts suggest complex sequence composition and potential ancient recombination.

(CS) score across the gene. For *Pf3D7_0420900*, the cumulative score strongly supports upsC. However, at the position corresponding to the tag (approximately amino acid position 184), there is a distinct dip in the cumulative score. The tag signal (green line) in this model is strongly associated with upsB, explaining the discrepancy in classification between the tag and full-length models. This suggests that the tag region is more similar to upsB, whereas the remainder of the gene aligns more closely with upsC.

A second example is shown in the bottom panel of Fig 4, where both the tag and full-length sequence of *Pf3D7_0412700* are classified as upsC. The cumulative sum of the positional weights score supports this, although the initial portion of the tag shows some signal associated with upsB, again suggesting a mixed origin.

These regional fluctuations in signal, where different tetrapeptides contribute positively or negatively to distinct classes, may be indicative of historical recombination events. The ability to track such regional contributions across *var* genes provides new insights into their complex evolutionary history.

## Application of upsML to global *P. falciparum* genome data

Finally, we applied upsML to the varDB dataset, which contains approximately 2,500 *P. falciparum* genomes. Most *var* gene sequences in this dataset are near-complete exon 1 sequences. We utilised the linear SVM model for 'exon 1' classification, as it is the most computationally efficient, capable of classifying 100 sequences in approximately 5–15 seconds, while maintaining comparable accuracy to more complex models.

Both classification models, the *ups* type and genomic location (internal vs subtelomeric), were applied. The resulting annotations have been uploaded to the *varDB* GitHub repository (https://github.com/ThomasDOtto/varDB/tree/master/Datasets/upsML).

To minimise biases arising from uneven sampling across regions, we analysed a normalised subset of the dataset consisting of 660 isolates from eleven countries (60 isolates per region), with each isolate containing between 52 and 65 *var* genes, except for the Senegal dataset (average of 48) (Table 6). While the primary goal of this analysis was to demonstrate the scalability and speed of upsML for rapid 'exon 1' annotation, taking less than one second to annotate 1,200 *var* genes, it also provides an overview of geographical patterns in ups types. In line with expectations, upsA and upsE were the most confidently classified types, but we did not observe clear geographical structuring for upsA.

In contrast, our positional classifier revealed a consistent enrichment of internal CB-type *var* genes in isolates from the Asia regions (Table 6, top). Frequencies were highest in Cambodia and Thailand (33.9% each) and lowest in Kenya and Mali (26.7% and 26.8%, respectively). To better understand those differences in all samples, we grouped the countries into East Africa (Kenya, Malawi, Congo – median 28.1%), West Africa (Senegal, The Gambia, Ghana, Mali – median 28.8%), and Asia (Thailand, Cambodia, Vietnam, Laos – median 32.6%, S1 Fig). Differences in internal *var* gene frequency were significant between Asia and the two African regions, p-values 3.4e-12 (West Africa) and 2.1e-09 (East Africa), but not within Africa (p-value = 0.04 – Welch Two Sample t-test). In contrast, we did not observe a similarly clear difference for upsC (Table 6, bottom), likely reflecting the known difficulty of distinguishing upsB from upsC based on sequence features alone. Thus, internal *var* genes are significantly more frequent in Asia.

While these observations highlight the utility of our positional classifier for large-scale comparative analyses, they must be interpreted with caution. The underlying varDB assemblies vary in sequencing technology, read length, sampling time, and assembly quality, which may differentially affect the recovery of specific *var* gene classes.

To draw further robust biological conclusions regarding *var* gene distributions, more controlled datasets processed under uniform protocols should be used. For example, large-scale datasets such as Pf8 [18] or RNA-Seq-derived contigs from clinical isolates stratified by disease phenotype [19] would provide a stronger basis for assessing associations between ups types, genomic location, and disease outcomes.

**Table 6. Classification of *var* genes from 660 isolates across eleven regions using upsML. The linear SVM model was applied to classify 'exon 1' protein sequences by genomic position (left) and ups type (right). Each region includes 60 isolates. Values represent the percentage of genes assigned to each class.**

| Country | Median #*var* | A | B subtelomeric | CB internal | E |
|---|---|---|---|---|---|
| Kenya | 52 | 27.9 | 45 | **26.7** | 1.7 |
| Mali | 65.5 | 22.2 | 47.5 | **26.8** | 1.8 |
| Congo | 62 | 17.9 | 51.3 | **27.8** | 1.8 |
| Malawi | 65 | 20.6 | 47.9 | **28.1** | 1.8 |
| The Gambia | 60 | 20.2 | 48.1 | **28.3** | 1.8 |
| Ghana | 65 | 20.5 | 47.4 | **30** | 1.6 |
| Vietnam | 56 | 24.5 | 40.4 | **31.3** | 1.9 |
| Senegal | 48 | 16.7 | 48.1 | **31.6** | 2.1 |
| Laos | 58.5 | 22.4 | 42.8 | **32.5** | 1.9 |
| Thailand | 54 | 19.9 | 44.7 | **33.9** | 1.9 |
| Cambodia | 56 | 20.5 | 43 | **33.9** | 1.9 |
| **Country** | **Median #var** | **A** | **B** | **C** | **E** |
| Mali | 65.5 | 21.2 | 64.9 | 9.6 | 1.8 |
| Kenya | 52 | 24.8 | 62.2 | 10.2 | 1.7 |
| Malawi | 65 | 20 | 66.4 | 10.3 | 1.8 |
| Congo | 62 | 17.6 | 69.1 | 10.9 | 1.8 |
| The Gambia | 60 | 19.7 | 66.3 | 11.4 | 1.8 |
| Ghana | 65 | 20.3 | 65.2 | 12.2 | 1.6 |
| Senegal | 48 | 16.3 | 68.1 | 12.2 | 2.1 |
| Cambodia | 56 | 19.7 | 63.2 | 12.7 | 1.9 |
| Vietnam | 56 | 24.6 | 60.4 | 12.7 | 1.9 |
| Thailand | 54 | 19.7 | 65.4 | 12.8 | 1.9 |
| Laos | 58.5 | 21.8 | 61.9 | 13 | 1.9 |

## Discussion

Understanding the ups type of *var* genes is valuable for interpreting their functional properties and roles in *Plasmodium falciparum* biology. However, obtaining upstream sequence information is often challenging. This is especially true for tag sequences or RNA-Seq data, where the upstream region may not be captured, and even in whole-genome Illumina datasets, low-complexity and repetitive features of ups sequences hinder assembly [20].

To address this, we developed upsML, a machine learning–based tool that predicts ups types based on sequence features. We evaluated multiple classifiers using different input types. This is a non-trivial task, as *var* genes are known to recombine frequently, especially within internal gene clusters [11,21,22]. Overall, larger k-mers generated better results, and the SVM models were slightly better than the other methods. Interestingly, the SVM model using only 1-mers already provides informative classification performance, suggesting that simple amino acid composition carries a signal for *ups* type discrimination. Future work using larger datasets could explore methods that incorporate positional information, such as recurrent neural networks or other deep learning approaches.

The tool performs well for upsA and upsE genes, which are distinct and less prone to recombination. In contrast, classification of upsB and upsC types is more challenging due to frequent recombination, particularly in internal clusters. Consequently, sensitivity and specificity for these types can be as low as 0.27. Because of class imbalance, a B call is correct

93% of the time, whereas 73% of C types are misclassified as B. These metrics improve with sequence length: overall accuracy rises from 0.82 (tags) to 0.90 ('exon 1') and 0.92 (full-length *Pf*EMP1 sequences). Interestingly, the inclusion of exon 2 shifts the sensitivity and specificity between upsB and upsC. While not formally tested, this suggests that exon 2 may be more tightly linked to the upstream region, potentially because recombination events are more likely to involve exon 1.

To further investigate classification errors, we applied an explainable AI approach by analysing the contribution of individual tetrapeptides to upsB or upsC classification in a *Pf3D7 var* gene that showed discordant results between that tag and *Pf*EMP1 sequences. For this gene, the tag sequence showed a strong upsB signal, whereas the remainder of the gene clearly leaned towards upsC. Interestingly, local drops in the cumulative classification signal aligned with potential recombination breakpoints, highlighting the mosaic structure of the gene. The mosaic nature of *var* genes has been frequently described [11,23,24], but our ability to observe it using tetrapeptides linked to upstream sequence types offers a novel approach to identifying sequence features associated with recombination events.

We also assessed whether increasing the amount of training data would improve performance. While this is likely, the recombinogenic nature of *var* genes imposes an inherent ceiling on classification accuracy. In comparing models, we found that performance was generally consistent across algorithms. However, for tag sequences, some models offered marginally higher accuracy at high computational cost – up to 100-fold slower than others. Given this, we opted for the fastest-performing model (linear SVM) for most input types and the polynomial SVM for tags. With trained models, annotating 1,200 *var* genes from 20 genomes takes approximately one second using linear SVM and 100 seconds using the polynomial model.

We benchmarked upsML against cUps, an HMM-based tool that utilises tag sequences. On their dataset (using leave-one-out cross-validation), upsML outperformed cUps by 10 percentage points (0.83 vs. 0.74 accuracy, Table 4). Notably, 17% of sequences in the cUps dataset were duplicates, suggesting the true accuracy is likely lower. When tested on our dataset, cUps' overall accuracy was lower, although it showed improved performance on upsC types. The predominance of upsB sequences per genome obscures this distinction.

Given the difficulty in distinguishing upsB and upsC, and the importance of chromosomal positioning in *var* gene biology, we developed an additional classifier to distinguish subtelomeric from internal *var* genes (Table 5). For tag sequences, accuracy reached 78%, and for full-length genes, up to 92%. This again suggests that exon 2 may be a strong indicator of chromosomal location [4]. Because most studies rely on tag sequences or partially assembled genomes, this tool enables reasonably accurate inference of genomic position in approximately 78–84% of cases. Applying this positional classifier to a larger global *var* gene dataset [20] (Table 6), we observed that the relative abundance of internally located *var* genes varies geographically. A major contrast between Asian and African parasite populations is transmission intensity, with higher transmission in central and eastern Africa and lower transmission in Asia. In low-transmission areas, parasites are expected to persist longer within individual hosts, favouring chronic infections. Nobel and colleagues [25] predicted that chronicity would be associated with increased expression of upsC (internal) *var* genes, and our analysis supports this prediction. On the other hand, because those numbers are relative, isolates in Africa may have more subtelomeric *var* genes, perhaps due to different clinical phenotypes or multiple infections, to maintain higher diversity. More broadly, we note that upsML can be used to generate novel hypotheses and help interpret epidemiological differences in *var* gene architecture.

More generally, despite the extreme polymorphism and recombination of *var* genes, our work demonstrates that machine learning methods can be applied effectively to classify these genes. Future work could focus on predicting functional features, such as domain cassettes, and linking these to clinical phenotypes.

In conclusion, upsML offers a robust and efficient tool for annotating *var* genes by their ups type and genomic location. It holds promise for enabling deeper insights into *var* gene expression patterns in natural infections and facilitating large-scale comparative studies.

## Methods

All custom programs were written in Python and can be found on our GitHub site (https://github.com/sii-scRNA-Seq/upsML). The models used in this analysis are also available on our GitHub.

### Preparation of var gene database

A database of *var* genes was extracted from long-read assemblies and used as training data for models. For this, 23 *Plasmodium falciparum* long read assemblies were recovered from literature and publicly available datasets [26–28]. Genome sequences were downloaded from NCBI, and assemblies were annotated using AUGUSTUS [29]. *Var* gene sequences were identified through a BLAST search on the list of protein-coding sequences. To this dataset, we added 2,835 *var* gene sequences from 43 isolates from The Gambia, which were extracted from high-quality PacBio HiFi long-read assemblies. The genomes were annotated through Companion v2 [30]. *Var* genes were retrieved from all assemblies by screening for genes annotated as potentially coding *Pf*EMP1 by the Companion pipeline and filtering for nucleotide lengths greater than 2500 bp. All identified *var* genes were passed through pre-established HMM models [4] to define their domain composition. Copies of *var2csa* were identified by extracting *var* genes that had a DBLpam domain. Those genes were then extracted from the dataset and annotated as upsE.

### Annotation of ups classifications

To annotate and link *var* genes to their upstream groups (ups), we used a combination of cluster-based analyses and phylogenetic trees. *Var* genes of known ups groups were taken from published datasets [4], and we mapped the ups annotations for those genes using SAMtools v1.18 [31] and blastn v2.14.0 [32] onto our sequences.

Then, to cluster all *var* genes into ups groups, we used phylogenetic tree clustering. First, the 500 bp upstream sequences of all *var* genes were extracted, then aligned with MAFFT v7.520 [33] and a phylogenetic tree was generated using FastTree 2v.1.11 [34]. The *var* genes of the known ups group were then used to identify and label phylogenetic clades in the tree. Sequences clustering with *var* genes representative of ups group A, B and C were assigned the corresponding annotation. To generate our final training set, we added the upsE sequences and excluded 3D7 reference sequences.

After assigning the ups types, we obtained a final set of 780 upsA, 2545 upsB, and 861 upsC *var* genes. Additionally, 110 *var2csa* sequences from the long read assemblies were then included in the dataset as upsE (S1 Table). To enhance the diversity found in the training datasets, highly similar sequences were filtered out. CD-HIT v4.8.1 [35] was used to cluster sequences at a 98% identity threshold within each gene section set. This retained the unique representatives with the longest sequence in each cluster. A final diverse dataset of 2530 *var* genes was obtained. This resulted in 468 upsA, 1,516 upsB, 507 upsC, and 39 upsE.

### Extraction of partial *var* sequences

To prepare training datasets for ML models, *var* genes were translated to protein sequences. We generated four types of datasets (Fig 1) by pulling out partial sequences corresponding to:

1. Tags: a sequence of ~150 amino acids located in the DBLα domain that can be amplified through degenerated PCR primers in field isolates. The N-terminal sequence always starts with "LARSFADIG" [36].

2. Cassettes defined the DBLα-CIDRα cassette pair.

3. 'Exon 1': A sequence that contains the NTS domain and is at least 3.5 kb long, but does not contain either a *var* intron nor the ATS domain.

4. Full-length *Pf*EMP1: The complete coding sequence of the *var* gene, including both exons.

 

This was performed through custom Python scripts.

Amongst the *var* genes, some sequences lacked one or more of these components, but to maximise the information gathered from these *var* genes and gather the largest training datasets, all identified gene sections were used. Additionally, *var* genes of the canonical Pf3D7 reference genome were removed from this dataset, as it was used for model verification. After excluding identical *var* genes, a final diverse dataset of 2,530 full-length *Pf*EMP1 sequences, 2,379 'exon 1's, 1,995 domain cassettes, and 1,970 tags was then used to train the machine learning models (S2 Table).

## Creating and testing machine learning models

To classify *var* gene sequences, amino acid sequences of each partial sequence type were decomposed into their raw peptides (k-mer = 1), dipeptides (k-mer = 2), and tetrapeptides (k-mer = 4) count tables. Those were then used as features for six different machine learning algorithms: Random Forest, XGBoost, and SVMs with four different kernels (Linear, polynomial, Radial Basis Function and Sigmoid). All implementations were done in Python; the scikit-learn library v1.3.2 [37] was used for all SVM and random forest models, while xgboost v2.1.1 was used for XGBoost. The hyperparameters of the models were optimised through the Python library optuna v4.1.0. The complete *var* genes datasets were split into 80% training, 20% test datasets and overall accuracies, precisions, sensitivities, and specificities were calculated and reported for each model using the Python libraries scikit-learn 1.3.2, Biopython 1.83, Pandas 2.0.3, and NumPy 1.24.4. [38].

## Comparison to cUps

To ensure a fair comparison with cUps, we first downloaded their original reference dataset of 846 DBLα tag sequences 11. We then applied leave-one-out cross-validation (LOOCV) using the same dataset on the upsML models. This allowed a direct comparison of classification accuracy under the same evaluation strategy reported in the original cUps study 11. We then trained it with the same input as upsML for a comparison with our data.

## Validation on reference 3D7 genome

We used the upsML models to annotate the ups type of the 61 3D7 *var* genes [39], with the tag sequences isolated from the full sequences using the previously mentioned method. The occurrence heatmap was done in Python with pandas 2.0.3, Seaborn 0.13.2, and Matplotlib 3.7.5.

## *Var* gene localisation model

Refined models with added information about *var* gene localisation were trained in a similar method. To do this, labels of the *var* sequences were changed to upsA, upsE, upsB_subtelomeric and upsBC_internal. We previously [27] manually annotated the chromosomal position of the *var* genes. For the remaining genes, we verify their position on chromosomes known to harbour internal *var* genes and annotate the *var* genes accordingly. From the 2,545 upsB dataset, 1,016 subtelomeric and 297 internal sequences were successfully identified. This resulted in an annotated dataset of 780 upsA, 110 upsE, 1,016 upsB_subtelomeric, and 1,158 upsBC_internal full-length *Pf*EMP1 sequences.

To produce the models using this alternate dataset, the gene regions were extracted as previously mentioned. This led to final unique datasets of 1,550 tags, 1,541 cassettes, 1,890 'exon 1"s, and 1,980 full-length *Pf*EMP1 sequences.

## Explainable AI

Feature importances were estimated for the tetrapeptide linear SVM model by fitting models that classify *var* sequences into the two classes, *var* group B/C or internal/external *var* genes. We used the weights of the SVM models, i.e., the coordinates of the vector orthogonal to the hyperplane separating the two classes [36], as direct measures of each feature's

(amino acid k-mer sequence) contribution to classify a PfEMP1 sequence into one class or the other. This feature vector was defined in 01_feature_importance.py by the Python calculate_tetrapeptide_counts function. Here is the pseudo code:

**Input:** sequence S
**Output**: tetrapeptide count vector V

1. If necessary, translate S to amino acid sequence.

2. Slide a window of size 4 along S to extract all overlapping tetrapeptides.

3. Count occurrences of each tetrapeptide.

4. Output counts in a fixed order defined by ALL_TETRAPEPTIDES.

The coef_ attribute of the scikit-learn LinearSVC class, defined in the documentation as the "weights assigned to the features", was used to extract weights from linear SVM models. In these code snippets from 01_feature_importance.py, clf.coef_ stores the coefficients of the linear model learned by LinearSVC:

**Input**: feature matrix X, label vector y, regularisation parameter C
**Output**: feature weight vector W

1. Initialise a linear Support Vector Machine classifier with parameter C.

2. Train the classifier on X and y.

3. Extract the learned feature weight vector W from the trained model.

4. Return W.

Using a sliding window, we then mapped the relative feature importances of each tetrapeptide along full PfEMP1 sequences. This allowed us to identify regions of PfEMP1 sequences associated with different classes (Fig 4a). By summing up the contributions of tetrapeptides along the protein sequences, we could also illustrate how tetrapeptide features progressively influenced the classification outcome from the start to the end of the sequence. The was done using the following function from 02_explainable_ai_plot.py that turns tetrapeptide SVM coefficients into a per-position "decision weight" signal across the sequence:

**Input**: protein sequence S, tetrapeptide weight mapping Wdict
**Output**: positional weight profile P along S

1. Convert sequence S to uppercase.

2. Initialise positional score array P of length equal to S with zeros.

3. For each position i from 1 to length(S) − 3:

   a. Extract tetrapeptide T = substring of S from i to i + 3.

   b. Retrieve weight w = Wdict[T]; if T is not present, set w = 0.

   c. Determine centre position c = i + 1 (central residue of tetrapeptide).

   d. Add w to P[c].

4. Return P.

We calculated the cumulative sum (CS) across the entire PfEMP1 sequences (red line in Figs 4b and 4c) to capture the overall classification bias implied by the linear SVM model along the protein. A subregional CS (green line in Fig 4b and 4c)

was also calculated to quantify the local contribution of features within shorter DBLa tag sequences. This calculation used the profile (the vector of per-position linear SVM decision weights) and the subregion (the amino-acid interval defining the DBLα tag). This was implemented in 02_explainable_ai_plot.py as:

**Input**: positional weight profile P along sequence S, subregion interval [a, b]

**Output**: full-sequence cumulative trace Cfull, subregional cumulative trace Csub

1. Compute cumulative sum of P over the full sequence to obtain Cfull.

2. Extract subregional profile segment Psub = P[a to b].

3. Compute cumulative sum of Psub to obtain Csub.

4. Return Cfull and Csub.

### 3K genomes

In 2018, we published the *var* genes of over 2500 field isolates [20]. To attribute their ups type, the full and normalised datasets of *var* genes as amino acids was downloaded from https://github.com/ThomasDOtto/varDB/tree/master/ Datasets/upsML. The upsML SVM linear models were used to attribute the ups and localisation types, with the results uploaded to varDB GitHub.

### Supporting information

**S1 Table. Complete Training Set of Full *Var* Genes and ups Assignments used.**
(PDF)

**S2 Table. All and Unique Training Sets of Isolated *Var* Genes Components Regions.**
(PDF)

**S3 Table. Overall accuracy of upsML, including shorter peptide models.**
(PDF)

**S4 Table. upsML Model Speeds when training then classifying 60 3D7 complete *Pf*EMP1 sequences.**
(PDF)

**S5 Table. Specificity and Sensitivity of Tetrapeptide Models for first model.** Sensitivity refers to the proportion of true positives, while specificity refers to the proportion of true negatives.
(PDF)

**S6 Table. Confusion Matrixes of Internal/ Subtelomeric Models.**
(PDF)

**S7 Table. Sensitivity and Specificity of Internal/Subtelomeric Models.** Split of results for the different sequence types.
(PDF)

**S1 Fig. Relative number of internal *var* genes per region.** Data from Table 6, of 660 field isolates, from 11 regions, classified with upsML. The genes were grouped as follows: Asia: Thailand, Cambodia, Vietnam and Laos (median 32.6); East Africa: Mali, Kenya, Malawi & Congo (median 28.1); West Africa: Senegal, The Gambia, Mali and Ghana (median 28.8). Isolates with fewer than 30 *var* genes in the assembly were ignored. The differences in abundance between Asia and the two African regions were significant, as indicated by a Welch Two-Sample t-test (both <1e-08).
(PDF)

## Author contributions

**Conceptualization:** Antoine Claessens, Thomas D Otto.

**Data curation:** Elcid Aaron Pangilinan, Mathieu Quenu.

**Formal analysis:** Elcid Aaron Pangilinan.

**Funding acquisition:** Thomas D Otto.

**Investigation:** Antoine Claessens.

**Methodology:** Elcid Aaron Pangilinan, Mathieu Quenu.

**Project administration:** Thomas D Otto.

**Software:** Elcid Aaron Pangilinan.

**Supervision:** Thomas D Otto.

**Writing – original draft:** Elcid Aaron Pangilinan, Thomas D Otto.

**Writing – review & editing:** Elcid Aaron Pangilinan, Mathieu Quenu, Antoine Claessens, Thomas D Otto.

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
