## [Decision Letter · Decision Letter 0]

8 Aug 2025

Dear Dr. Otto,

Thank you for submitting your manuscript to PLOS ONE. After careful consideration, we feel that it has merit but does not fully meet PLOS ONE’s publication criteria as it currently stands. Therefore, we invite you to submit a revised version of the manuscript that addresses the points raised during the review process.

We look forward to receiving your revised manuscript.

Kind regards,

Kokouvi Kassegne

Academic Editor

PLOS ONE

Journal Requirements:

2. Thank you for stating the following financial disclosure: [TDO was supported by the ExposUM Institute of the University of Montpellier (grants ANR-21-EXES-0005 and Occitanie Region) and the Wellcome Trust: 104111/Z/14/Z & A. EAP was funded by La Caixa Foundation - Health Research Program (HR20-00635).].

Please state what role the funders took in the study.  If the funders had no role, please state: """"The funders had no role in study design, data collection and analysis, decision to publish, or preparation of the manuscript.""""

Additional Editor Comments:

Please provide point-by-point response to the reviewers' comments and revise the manuscript accordingly to show where revisions have been made.

Reviewers' comments:

Reviewer's Responses to Questions

**Comments to the Author**

1. Is the manuscript technically sound, and do the data support the conclusions?

Reviewer #1: Yes

Reviewer #2: Yes

Reviewer #3: Yes

2. Has the statistical analysis been performed appropriately and rigorously?

Reviewer #1: Yes

Reviewer #2: Yes

Reviewer #3: Yes

3. Have the authors made all data underlying the findings in their manuscript fully available?

Reviewer #1: Yes

Reviewer #2: Yes

Reviewer #3: Yes

4. Is the manuscript presented in an intelligible fashion and written in standard English?

Reviewer #1: Yes

Reviewer #2: Yes

Reviewer #3: Yes

Reviewer #1: Pangilinan et al present a new tool for predicting the upstream region classification of var genes encoding P .falciparum’s key virulence proteins , PfEMP1). This is important as the var genes, despite their extreme diversity generated though both recombination and mutation, are well organized in the genome with specific receptor-binding phenotypes linked to specific chromosomal locations and upstream regions. This has led reserachers to use these sequence signatures to link receptor binding phenotypes to clinical outcome, and speculate if also genetic regulation may be tied to e.g. parasites establishing low patent chronic “dry season “infections or fast switching at the eary onset of infection etc. The present tool for linking ups-type to gene fragments which can be extracted experimentally- most commonly and DBLa tags is a welcoming tool, not least as an earlier attempt for such a tool was considered highly inaccurate – a notion supported by the present analyses.

A main challenge is within the biological limitations provided by the recombinogenic nature of the var genes.. ie the “truth” is that most likely not all var genes can be classified as either upsa, b or c – due to their recent recombination between the groups. As In addition to the upsABC the authors attempt to predict the chromosomal location from encoded sequence fragments, - again this may be inherently difficult for recently recombined genes.

On this background the authors apply and compare three different computational approaches to tackle the classification. The reporting on this is sound hones and open about the limitations – which - as expected and supported by 2 examples of in-depth analysis of ups-type calling sequence elements tiled across the var sequence, - appear to reflect said biological limitations.

It is this open and clear description of the limitations of the too, which makes the strength of the paper and will guide proper applications and data interpretations from the tool. I think this is well done.

A key application will be the annotation of expressed DBLa tags as these are the most easily acquired through NGS. A challenge for using the tool is that it relies on protein sequence. Have the authors tried to apply raw error prone DBLa tag DNA sequences ? if this was possible this would omit need for cumbersome data curation of large DBLa tag datasets?

In my view the use of DC is less likely to happen or be meaningful. The mention of domain cassettes (DCs) line 69 to 71, and throughout MS could be omitted as these have proven inaccurate and obsolete descriptions of both pfemp1 geno- and phenotypes. If kept i suggest some subtle changes below:

Line 73: For instance, half of group A and some B variants (DC8) bind …, whereas group A and C variants bind CD36 and are… (ie remove “tend to”).

Similarly the commonly used phrase "such as" on PfEMP1 endothelial receptors should changed as 1) it implies there is something grouping and common about CD36 EPCR and ICAM1 among endothelial receptors- there is not to my knowledge..2) also it implies some unspecific binding to a panel of receptors within ec receptors - this is not the case either..PfEMP1 have shown to be highly adapted to their human receptors.

I suggest "via specific binding to host receptors including cd36, EPCR and ICAm1..."

Line 64: strictly speaking: var2csa is not conserved ....

Section from line 156: The use of "var" gene sequence regions is misleading as what was used is the translated pfemp1 (protein ) sequences.

Reviewer #2: The authors address an important issue, the ability to infer further information about var gene sequences based on the partial sequences typically obtained from volunteer samples. This has value for evolutionary and genetics perspectives, attempting to understand the recombinations that underly the phylogeny of these gene family, and also may prove useful in identifying associations between var sequences and clinical phenotypes. Associations between ups types and severe disease have been frequently noted but remain unexplained, a robust means of predicting characteristics of these genes in the absence of the definitive upstream sequence data will allow such associations to be tested in larger datasets. The ultimate goal would be to predict the set of genes to which a tag or contig could belong and this work is a valuable step along that path.

The developed tool has advantages and disadvantages compared to the already available cUPS namely upsAI classifies ups B better and cUPS classifies ups C better, this appears to be because upsAI claims more Cs as Bs and cUPS claims more Bs as Cs. As the authors note this difference affects accuracy due to the greater number of B than C genes in the genome. Combining the two approaches would enable a relatively stringent set of identifications to be made but would leave a large number of genes as either upsC or B.

The authors made the interesting biological observations that he tools worked best on models that segregated genes by their chromosome position rather than by ups B/C. This suggests that constrained recombination due to genomic location has segregated the coding sequences for chromosomal internal upsB despite the homology between upsB sequences from telomeres or internal sites. Their tag analysis further substantiates this showing that tags can be associated with another ups type than the actual gene they belong to, presumably due to recombination.

Overall this paper makes a useful contribution providing a novel tool for var gene classification and using it to make several interesting observations about var gene recombination that inform our understanding of the gene family’s evolution.

I have only 2 minor suggestions,

Line 22 “the best model of upsAI for DBLα-tags sequences achieves an overall accuracy of 83%, 92% and for full-length PfEMP1 sequences, therefore significantly outperforming existing tools.”

There appears to be a typo in this sentence

Lines 388-391

“However, we advise caution in interpreting these patterns, as the underlying dataset is subject to biases related to sequencing quality and assembly methods. To draw robust biological conclusions regarding var gene distributions, more controlled datasets, such as RNA-Seq-derived contigs from clinical isolates stratified by disease phenotype, would be necessary. “

Why would rnaseq be more informative about var distributions? Presumably not all the var repertoire would be expressed and there are more likely to issues with uneven 5' coverage from rnaseq. Clinical rnaseq would allow associations between ups type and clinical disease to be identfied. Could the authors clarify what they mean by this statement.

Reviewer #3: Please see attached review.

**Do you want your identity to be public for this peer review?** For information about this choice, including consent withdrawal, please see our For information about this choice, including consent withdrawal, please see our Privacy Policy .

Reviewer #1: No

Reviewer #2: No

Reviewer #3: **Yes:** Yao-ban ChanYao-ban Chan

While revising your submission, please upload your figure files to the Preflight Analysis and Conversion Engine (PACE) digital diagnostic tool, https://pacev2.apexcovantage.com/ . PACE helps ensure that figures meet PLOS requirements. To use PACE, you must first register as a user. Registration is free. Then, login and navigate to the UPLOAD tab, where you will find detailed instructions on how to use the tool. If you encounter any issues or have any questions when using PACE, please email PLOS at . PACE helps ensure that figures meet PLOS requirements. To use PACE, you must first register as a user. Registration is free. Then, login and navigate to the UPLOAD tab, where you will find detailed instructions on how to use the tool. If you encounter any issues or have any questions when using PACE, please email PLOS at figures@plos.org . Please note that Supporting Information files do not need this step.. Please note that Supporting Information files do not need this step.

---

## [Author Response · Author response to Decision Letter 1]

30 Sep 2025

Reviewer #1:

Pangilinan et al present a new tool for predicting the upstream region classification of var genes encoding P .falciparum’s key virulence proteins , PfEMP1). This is important as the var genes, despite their extreme diversity generated though both recombination and mutation, are well organized in the genome with specific receptor-binding phenotypes linked to specific chromosomal locations and upstream regions. This has led reserachers to use these sequence signatures to link receptor binding phenotypes to clinical outcome, and speculate if also genetic regulation may be tied to e.g. parasites establishing low patent chronic “dry season “infections or fast switching at the eary onset of infection etc. The present tool for linking ups-type to gene fragments which can be extracted experimentally- most commonly and DBLa tags is a welcoming tool, not least as an earlier attempt for such a tool was considered highly inaccurate – a notion supported by the present analyses.

A main challenge is within the biological limitations provided by the recombinogenic nature of the var genes.. ie the “truth” is that most likely not all var genes can be classified as either upsa, b or c – due to their recent recombination between the groups. As In addition to the upsABC the authors attempt to predict the chromosomal location from encoded sequence fragments, - again this may be inherently difficult for recently recombined genes.

On this background the authors apply and compare three different computational approaches to tackle the classification. The reporting on this is sound hones and open about the limitations – which - as expected and supported by 2 examples of in-depth analysis of ups-type calling sequence elements tiled across the var sequence, - appear to reflect said biological limitations.

It is this open and clear description of the limitations of the too, which makes the strength of the paper and will guide proper applications and data interpretations from the tool. I think this is well done.

Answer:

We would like to thank the reviewer for their throughtful and positive reply.

A key application will be the annotation of expressed DBLa tags as these are the most easily acquired through NGS. A challenge for using the tool is that it relies on protein sequence. Have the authors tried to apply raw error prone DBLa tag DNA sequences ? if this was possible this would omit need for cumbersome data curation of large DBLa tag datasets?

Answer:

We appreciate this important suggestion. At present, upsML requires error-free translated protein sequences as input. Our tests with raw, error-prone DBLa tag DNA sequences confirmed that upsML does not perform reliably without prior error correction or translation.

We agree that developing methods to handle raw sequences directly would greatly simplify the workflow. Indeed, we are exploring a separate pre-processing pipeline to translate and error-correct DBLa tags prior to annotation, but we view this as a complementary tool rather than a core component of upsML.

In my view the use of DC is less likely to happen or be meaningful. The mention of domain cassettes (DCs) line 69 to 71, and throughout MS could be omitted as these have proven inaccurate and obsolete descriptions of both pfemp1 geno- and phenotypes. If kept i suggest some subtle changes below:

Line 73: For instance, half of group A and some B variants (DC8) bind …, whereas group A and C variants bind CD36 and are… (ie remove “tend to”).

Answer:

We appreciate the reviewer’s perspective regarding the validity of domain cassettes (DCs). We recognise that there has been debate about their accuracy; however, many recent studies on var gene expression and pathogenesis continue to use DC classification as a point of reference. For this reason, we believe it remains useful to include this terminology for readers who may encounter it in the literature.

As suggested, we have revised line 73 to read:

“For instance, group A and some B variants (DC8) bind to EPCR and are associated with cerebral malaria, whereas group B and C variants bind to CD36 and are linked to milder infections “

Similarly the commonly used phrase "such as" on PfEMP1 endothelial receptors should changed as 1) it implies there is something grouping and common about CD36 EPCR and ICAM1 among endothelial receptors- there is not to my knowledge..2) also it implies some unspecific binding to a panel of receptors within ec receptors - this is not the case either..PfEMP1 have shown to be highly adapted to their human receptors.

I suggest "via specific binding to host receptors including cd36, EPCR and ICAm1..."

Answer:

Thank you for highlighting this point and for suggesting clearer wording. We agree that “such as” could be misleading in this context. The introduction has been modified as suggested (lines 50–51) and now reads:

PfEMP1 enables iRBCs to adhere to endothelial cells via specific binding to host receptors including CD36, EPCR, or ICAM1, allowing parasites to avoid splenic clearance but contributing to microvascular obstruction and inflammation, hallmarks of malaria pathology

Line 64: strictly speaking: var2csa is not conserved ....

Answer:

The line has been modified to ‘nearly conserved sequence’ and now reads:

" Group E comprises the unusual var gene named var2csa, which has a nearly conserved sequence.”

Section from line 156: The use of "var" gene sequence regions is misleading as what was used is the translated pfemp1 (protein ) sequences.

Answer:

Yes, we agree. We modified it to ‘PfEMP1’ and is now:

“To predict upstream types from PfEMP1 sequences, we tested Support Vector Machines (SVMs), Random Forest (RanFor), and XGBoost (XGB) across four categories of partial protein sequence types...”

Reviewer #2:

The authors address an important issue, the ability to infer further information about var gene sequences based on the partial sequences typically obtained from volunteer samples. This has value for evolutionary and genetics perspectives, attempting to understand the recombinations that underly the phylogeny of these gene family, and also may prove useful in identifying associations between var sequences and clinical phenotypes. Associations between ups types and severe disease have been frequently noted but remain unexplained, a robust means of predicting characteristics of these genes in the absence of the definitive upstream sequence data will allow such associations to be tested in larger datasets. The ultimate goal would be to predict the set of genes to which a tag or contig could belong and this work is a valuable step along that path.

The developed tool has advantages and disadvantages compared to the already available cUPS namely upsAI classifies ups B better and cUPS classifies ups C better, this appears to be because upsAI claims more Cs as Bs and cUPS claims more Bs as Cs. As the authors note this difference affects accuracy due to the greater number of B than C genes in the genome. Combining the two approaches would enable a relatively stringent set of identifications to be made but would leave a large number of genes as either upsC or B.

The authors made the interesting biological observations that he tools worked best on models that segregated genes by their chromosome position rather than by ups B/C. This suggests that constrained recombination due to genomic location has segregated the coding sequences for chromosomal internal upsB despite the homology between upsB sequences from telomeres or internal sites. Their tag analysis further substantiates this showing that tags can be associated with another ups type than the actual gene they belong to, presumably due to recombination.

Overall this paper makes a useful contribution providing a novel tool for var gene classification and using it to make several interesting observations about var gene recombination that inform our understanding of the gene family’s evolution.

Answer:

We would like to thank the reviewer for their feedback and positive evaluation. We agree that for tags, Cups and upsML could be combined, however, the advantage of upsML is that it is designed and tested on different input types.

I have only 2 minor suggestions,

Line 22 “the best model of upsAI for DBLα-tags sequences achieves an overall accuracy of 83%, 92% and for full-length PfEMP1 sequences, therefore significantly outperforming existing tools.”

There appears to be a typo in this sentence

Answer:

Thank you for spotting this, we modified it to:

“The best model of upsML for DBLα-tags sequences achieves an overall accuracy of 83%, and 92% for full-length PfEMP1 sequences...”

Lines 388-391

“However, we advise caution in interpreting these patterns, as the underlying dataset is subject to biases related to sequencing quality and assembly methods. To draw robust biological conclusions regarding var gene distributions, more controlled datasets, such as RNA-Seq-derived contigs from clinical isolates stratified by disease phenotype, would be necessary. “

Why would rnaseq be more informative about var distributions? Presumably not all the var repertoire would be expressed and there are more likely to issues with uneven 5' coverage from rnaseq. Clinical rnaseq would allow associations between ups type and clinical disease to be identfied. Could the authors clarify what they mean by this statement.

Answer:

Thank you for pointing this out. We agree that our original wording was unclear. Our intention was not to imply that RNA-Seq would provide a complete picture of var gene distributions across the entire repertoire. Rather, we meant that RNA-Seq datasets from clinical isolates—especially when processed with consistent protocols and stratified by disease phenotype—could enable associations between ups types and clinical outcomes to be explored in a controlled manner.

To clarify, we have revised the text as follows:

“However, we advise caution in interpreting these patterns, as the underlying dataset is subject to biases related to sequencing quality and assembly methods. To draw robust biological conclusions regarding var gene distributions, more controlled datasets processed under uniform protocols should be used. For example, large-scale datasets such as Pf7, or RNA-Seq-derived contigs from clinical isolates stratified by disease phenotype, could enable more reliable associations between ups types and clinical outcomes to be identified.”

Reviewer #3: from attached PDF.

In this paper, the authors develop a new machine learning method (based on support

vector machines) to classify the upstream sequences of the var genes of Plasmodium falciparum, based on the protein sequences (or subsequences) of the gene itself. They show that their method is able to successfully classify with high accuracy, including on the upsB/C divide which has previously been difficult to separate, and outperforms current methods in the literature.

The paper is well-written and easy to read, and the problem it tackles is interesting and

topical. The results are good, and represent a significant advance on the current state-of-the-art. There are some issues that I would like to see addressed, and areas that I would like to see explored a bit further, but overall I think it is a nice addition to the literature.

Major comments:

1. The characterisation of the method as an “AI” method is inaccurate. The methods

used here are well-known and well-established machine learning methods, but they are

not AI methods in the modern sense of deep learning/neural networks. Having the

term “AI” in the title and in the name of the method is misleading at best; when I

read the title, I was indeed expecting some kind of deep neural network. To further

characterise the method as “explainable AI”, when this is unrelated to the modern

(and topical) sense of extracting meaning from “black box”-style models, seems to me

to be a little disingenuous.

Answer:

This is indeed a very valid point, and we renamed the tools throughout the manuscript, GitHub and on the command line.

2. Why were the models only trained with 1-, 2-, and 4-mers? The improving performance

with respect to length suggests that longer lengths should have been tried. I understand

that at some point there are too many feature dimensions, but certainly 3-mers (at the

very least) should have been tried.

Answer:

Thank you for this suggestion. Our initial choice of 1-, 2-, and 4-mers was guided primarily by computational efficiency, as the feature space grows exponentially with increasing k-mer length and models with k ≥ 5 became computationally impractical for our datasets.

In response to the reviewer’s comment, we re-ran the analyses using the fastest model (SVM Linear) with 3-mers included. As expected, performance improved with longer k-mers: 3-mers outperformed 2-mers, and 4-mers outperformed 3-mers, although the performance gains between successive k-mer lengths were modest (see Table below).

Tripeptied (k=3) Tetrapeptied (k=4)

tag 0.8350 0.8350

cassette 0.8421 0.8697

‘exon1’ 0.8845 0.8950

PfEMP1 0.9111 0.9190

3. It is somewhat improper to attempt several different machine learning methods and

report the best classification accuracy among them as the final numerical answer.

Due to random variation, one of the methods will always be slightly better than the

others, but not systematically. Indeed, this is exactly what we see in (e.g.) Table

11, where there is (to my eye) no significant difference between the methods, but the

best number among them is being claimed as the “overall accuracy”. This leads to

different methods being used for different input data types, for no biological reason.

What should be done is that a single method should be chosen based on this Table

(probably the linear SVM), and the accuracy of that one method reported from then

on.

Answer:

We appreciate the reviewer’s point regarding the interpretation of classification accuracy across multiple methods. Our intention in presenting results from several machine learning methods was to provide users with a suite of approaches rather than to claim superiority of a single model. While we agree that the observed differences in overall accuracy are small, we included multiple methods to highlight potential trade-offs in sensitivity and specificity across different sequence types (e.g., tag sequences, domains, exon 1, and full-length genes).

Because upsML allows users to analyse a variety of input data types, we believe it is informative to report the performance of multiple methods so that users can select the most appropriate approach for their specific dataset and research question. For this reason, we would prefer to keep the representation as is while clarifying in the text that the differences in overall accuracy are modest and not the main focus of our comparisons.

4. l289: I would not characterise this as a “classifier for genome location”; I would expect

such a thing to predict the actual position in the genome. What is being done here is

rearranging the upsB/C divide according to genome location. I think this is interesting

enough in its own right without overselling it. Likewise, I would consider l325–326 as

somewhat overselling the ability to predict chromosomal position.

Answer:

We appreciate the reviewer’s point regarding the terminology used for this analysis. Our goal was not to imply prediction of absolute chromosomal coordinates but rather to classify var genes into the well-established subtelomeric versus internal genomic locations. Since the Plasmodium falciparum genome was first published in 2003, these two locations have been recognised as biologically meaningful, with recombination rates and phenotypic associations differing by location.

Given the difficulty in distinguishing upsB and upsC groups, we felt it was important to highlight this classification explicitly, as it provides a novel way to integrate chromosomal positioning into var gene analyses. To clarify this point and avoid any sense of overstatement, we have revised the discussion as follows:

“Given the difficulty in distinguishing upsB and upsC, and the interest in chromosomal positioning of var genes, we developed an additional classifier to disti

---

## [Decision Letter · Decision Letter 1]

10 Nov 2025

Dear Dr. Otto

Thank you for submitting your manuscript to PLOS ONE. After careful consideration, we feel that it has merit but does not fully meet PLOS ONE’s publication criteria as it currently stands. Therefore, we invite you to submit a revised version of the manuscript that addresses the points raised during the review process.

We look forward to receiving your revised manuscript.

Kind regards,

Kokouvi Kassegne

Academic Editor

PLOS ONE

Journal Requirements:

Additional Editor Comments :

Please provide point-by-point to to the reviewer's minor comments and revise the manuscript accordingly to show (in tracking change mode) where revisions are made.

Reviewers' comments:

Reviewer's Responses to Questions

**Comments to the Author**

Reviewer #1: All comments have been addressed

Reviewer #2: All comments have been addressed

Reviewer #3: (No Response)

2. Is the manuscript technically sound, and do the data support the conclusions?

Reviewer #1: Yes

Reviewer #2: (No Response)

Reviewer #3: Yes

3. Has the statistical analysis been performed appropriately and rigorously?

Reviewer #1: Yes

Reviewer #2: (No Response)

Reviewer #3: Yes

4. Have the authors made all data underlying the findings in their manuscript fully available?

Reviewer #1: Yes

Reviewer #2: (No Response)

Reviewer #3: Yes

5. Is the manuscript presented in an intelligible fashion and written in standard English?

Reviewer #1: Yes

Reviewer #2: (No Response)

Reviewer #3: Yes

Reviewer #1: The authors have addressed all my comments appropriately and made changes to the manuscript accordingly.

Reviewer #2: (No Response)

Reviewer #3: Please see attached pdf.

I include some filler words here to attain the lower character limit. (This is a rather absurd requirement.)

**Do you want your identity to be public for this peer review?** For information about this choice, including consent withdrawal, please see our For information about this choice, including consent withdrawal, please see our Privacy Policy .

Reviewer #1: No

Reviewer #2: No

Reviewer #3: No

---

## [Author Response · Author response to Decision Letter 2]

12 Dec 2025

This is already in the submitted file, but we will paste it in again

Reviewer 3:

In this paper, the authors develop a new machine learning method (based on support

vector machines) to classify the upstream sequences of the var genes of Plasmodium falci-

parum, based on the protein sequences (or subsequences) of the gene itself. They show that

their method is able to successfully classify with high accuracy, including on the upsB/C

divide which has previously been difficult to separate, and outperforms current methods in

the literature.

Although the authors have made a credible attempt to address my points, and have

mostly done so successfully, there remain some points which I am not completely convinced

about. I go through these below, including my original review and the authors’ responses in

italics.

Major comments:

1. It is somewhat improper to attempt several different machine learning methods and

report the best classification accuracy among them as the final numerical answer.

Due to random variation, one of the methods will always be slightly better than the

others, but not systematically. Indeed, this is exactly what we see in (e.g.) Table

1, where there is (to my eye) no significant difference between the methods, but the

best number among them is being claimed as the “overall accuracy”. This leads to

different methods being used for different input data types, for no biological reason.

What should be done is that a single method should be chosen based on this Table

(probably the linear SVM), and the accuracy of that one method reported from then

on.

We appreciate the reviewer’s point regarding the interpretation of classification accu-

racy across multiple methods. Our intention in presenting results from several machine

learning methods was to provide users with a suite of approaches rather than to claim

superiority of a single model. While we agree that the observed differences in overall

accuracy are small, we included multiple methods to highlight potential trade-offs in

sensitivity and specificity across different sequence types (e.g., tag sequences, domains,

exon 1, and full-length genes).

Because upsML allows users to analyse a variety of input data types, we believe it

is informative to report the performance of multiple methods so that users can select

the most appropriate approach for their specific dataset and research question. For

this reason, we would prefer to keep the representation as is while clarifying in the

text that the differences in overall accuracy are modest and not the main focus of our

comparisons.

I’m afraid I am not really convinced by this argument. The problem here is that there

is no biological distinction between the various approaches; rather the distinction is

statistical, through the use of different kernel functions. If there were credible reasons

that one of these kernel functions would work well for a certain data type and a different

one would work well for another type, then it would make sense to present multiple

approaches. However, as I observed earlier, the variation between the methods appears

to be largely random noise, so that different approaches work better for different types

simply because there must be a best method. Biologically, there is no reason that one

should use e.g., a polynomial kernel for tags but a linear kernel for cassettes. From an

end-user perspective, the approaches are largely interchangeable, and it is difficult to

resolve conflicting outcomes. I think one should not resolve these differences based on

differences in classification accuracy of 0.01.

Response:

We agree that we might have been focusing on small differences here. However, we do think it is legitimate to test different methods and understand sensitivity and specificity. Hence, we change the message of the paper as requested.

For example in the abstract we write:

Several models in upsML achieve an accuracy of 83% for DBLα-tags sequences and 92% for full-length PfEMP1 sequences. – rather than the best method.

Next in the results we wrote:

The differences between models within each sequence type were small (1–2%), which were not statistically significant.

Several models perform similarly; therefore, we recommend using the one with the lowest runtime.

We do think that it is interesting for the reader to under the differences between the models for the false positive (Table 2) and Fig 3 – the Pf3D7 test.

And finally, we already wrote:

However, for tag sequences, some models offered marginally higher accuracy at significant computational cost - up to 100-fold slower than others. Given this, we opted for the fastest-performing model (linear SVM) for most input types and the polynomial SVM for tags. With trained models, annotating 1,200 var genes from 20 genomes takes approximately one second using linear SVM and 100 seconds using the polynomial model

So we think that with the changes, we make it clear that the approaches are very similar and we suggest to use the one that has a shorter runtime.

Comment 2.

Figure 4: In general I like this idea a lot and I think it clearly shows the inclination

of various subsequences towards the different ups types, which to my knowledge has

not been studied before. However, the details of the method used here need to be

expanded; the Methods section (l539) does not give the mathematical formulas used. I

can guess what is going on, but I should not have to. In the Results section, the AUC

score is not defined in l339. I was hoping that it would be defined later, but when I

read the relevant Methods section, there does not appear to be any mention of AUC.

First, we agree that we have to include the calculation of the AUC into the methods,

which we did. The calculation of feature importance is explained under “Creating and

Testing Machine Learning Models”: (text removed for brevity)

And later the calculation of AUC as used in Figure 4 is now covered in “Explainable

AI”: (text removed for brevity)

The Methods section is still not detailed enough for me to fully understand what is

going on. (For context, I am a mathematician, and I would like to know the mathemat-

ical details used here.) There is a repeated reference to the ‘weight’ of a tetrapeptide

— how is that defined and extracted from the SVM model? The “Regional Decision

Weight” is not clearly defined — for example, how large are the windows used? I also

question if it makes sense to sum Regional Decision Weights along the sequence, since

they are already local aggregates in a window. Similarly, what curve is the AUC the

area under of? The traditional formulation of AUC as the area under an ROC or

sensitivity/specificity curve doesn’t seem to apply here. What I would really like to

see are concrete mathematical formulas that enable the reader to exactly identify and

reproduced what is being calculated, and I think the authors have not gone far enough

in this respect.

ANSWER:

We now understand the detail the referee wants to see. From line 581, we described in more details what was done. Especially, we define the “weights” and the windows size. We chose to describe this as the code we used, as it would make it easier for the readership than to read mathematical descriptions. We hope that the editor and reviewer agree here.

Comment 3. l379: There should be more analysis of the results here than a simple table. The

text considers that it is “informative to observe geographical variation”, but what

information should be observed here?

ANSWER:

We wrote know down the interesting information, that the relative frequency of internal var genes is high in Asia.

We added the text in three part of the paper.

Abstract

Additionally, we developed a model to distinguish internal from subtelomeric var genes, which we applied to a global collection of P. falciparum genomes, revealing a higher frequency of internal var genes in Asia.

Results

In contrast, our positional classifier revealed a consistent enrichment of internal CB-type var genes in isolates from Asia regions (Table 6, top). Frequencies were highest in Cambodia and Thailand (33.9% each) and lowest in Kenya and Mali (26.7% and 26.8%, respectively). To better understand those differences in all samples, we grouped the countries into East Africa (Kenya, Malawi, Congo – median 28.1%), West Africa (Senegal, The Gambia, Ghana, Mali – median 28.8%), and Asia (Thailand, Cambodia, Vietnam, Laos – median 32.6 %, Figure S1). Differences in internal var gene frequency were significant between Asia and the two African regions, p-values 3.4e-12 (West Africa) and 2.1e-09 (East Africa), but not within Africa (p-value = 0.04). In contrast, we did not observe a similarly clear difference for upsC (Table 6, bottom), likely reflecting the known difficulty of distinguishing upsB from upsC based on sequence features alone. Thus, internal var genes are significantly more frequent in Asia.

Discussion

Applying this positional classifier to a larger global var gene dataset [20] (Table 6), we observed that the relative abundance of internally located var genes varies geographically. A major contrast between Asian and African parasite populations is transmission intensity, with higher transmission in central and eastern Africa and lower transmission in Asia. In low-transmission areas, parasites are expected to persist longer within individual hosts, favouring chronic infections. Nobel and colleagues [25] predicted that chronicity would be associated with increased expression of upsC (internal) var genes, and our analysis supports this prediction. On the other hand, because those numbers are relative, isolates in Africa may have more subtelomeric var genes, perhaps due to different clinical phenotypes or multiple infections, to maintain higher diversity. More broadly, we note that upsML can be used to generate novel hypotheses and help interpret epidemiological differences in var gene architecture.

Minor comments:

Thank you for spotting those issues in the text.

• l140: “mapping of all of them”

We are not sure what the reviewer means here:

Line 139 - 141 Sequences annotated as upsE, corresponding to the

conserved var2csa, were excluded from phylogenetic-based clustering due to their distinct and

well-characterised nature.

• Table 3: The table is now too wide for the page.

Fixed, but I hope that the Plos One type setter will also have a look here.

• Table 4: The table now appears to be duplicated, and the first part of the second table

looks a bit weird.

Fixed

• Table 5: The caption “Internal/telomeric” is not really in a sensible place.

Fixed

• l398: Doubled sentence

Fixed.

• l421: Although I take the meaning, GC content is not a property of amino acid com-

position.

Yes, stupid. Now reads:

simple amino acid sequence composition features carry signal

• l532: “the tetrapeptide linear SVM model”

fixed

• l562: Since all other instances of “AI” have been removed from the text, this heading

no longer seems appropriate. Likewise the keyword (l47).

Although we agree that the classifiers in upsML are not “AI” in the broad sense, the interpretability analyses we apply do fall within established definitions of Explainable AI (XAI). Our use of linear SVM coefficients for feature importance, the derivation of local and global explanatory profiles, and the regional model diagnostics all correspond to standard XAI methods. For this reason, we feel the heading remains appropriate, and we have clarified these steps in the revised Methods.

• l592: “mentioned in the Methods”.

We agree, but I think we need to keep it here explicitly as format of Plos one.

• l598: Heading formatting

Fixed

---

## [Decision Letter · Decision Letter 2]

18 Jan 2026

Dear Dr. Thomas D Otto,

Thank you for submitting your manuscript to PLOS ONE. After careful consideration, we feel that it has merit but does not fully meet PLOS ONE’s publication criteria as it currently stands. Therefore, we invite you to submit a revised version of the manuscript that addresses the points raised during the review process.

We look forward to receiving your revised manuscript.

Kind regards,

Kokouvi Kassegne

Academic Editor

PLOS One

Journal Requirements:

Additional Editor Comments:

Minor revisions are requested; please refer to the attachment

Reviewers' comments:

Reviewer's Responses to Questions

**Comments to the Author**

Reviewer #3: (No Response)

2. Is the manuscript technically sound, and do the data support the conclusions?

Reviewer #3: Yes

3. Has the statistical analysis been performed appropriately and rigorously?

Reviewer #3: Yes

4. Have the authors made all data underlying the findings in their manuscript fully available?

Reviewer #3: Yes

5. Is the manuscript presented in an intelligible fashion and written in standard English?

Reviewer #3: Yes

Reviewer #3: Please see attached review.

**Do you want your identity to be public for this peer review?** For information about this choice, including consent withdrawal, please see our For information about this choice, including consent withdrawal, please see our Privacy Policy .

Reviewer #3: No

---

## [Author Response · Author response to Decision Letter 3]

26 Jan 2026

This is also attached as work document, but also pasted here:

Reviewer 3

In this paper, the authors develop a new machine learning method (based on support

vector machines) to classify the upstream sequences of the var genes of Plasmodium falci-

parum, based on the protein sequences (or subsequences) of the gene itself. They show that

their method is able to successfully classify with high accuracy, including on the upsB/C

divide which has previously been difficult to separate, and outperforms current methods in

the literature.

The authors have responded well to the points raised in my latest review. With one

exception, I am happy with the changes and any remaining comments are minor.

Major comment (original comment and response first):

1. The Methods section is still not detailed enough for me to fully understand what is

going on. (For context, I am a mathematician, and I would like to know the mathemat-

ical details used here.) There is a repeated reference to the ‘weight’ of a tetrapeptide

— how is that defined and extracted from the SVM model? The “Regional Decision

Weight” is not clearly defined — for example, how large are the windows used? I also

question if it makes sense to sum Regional Decision Weights along the sequence, since

they are already local aggregates in a window. Similarly, what curve is the AUC the

area under of? The traditional formulation of AUC as the area under an ROC or

sensitivity/specificity curve doesn’t seem to apply here. What I would really like to

see are concrete mathematical formulas that enable the reader to exactly identify and

reproduced what is being calculated, and I think the authors have not gone far enough

in this respect.

Answer:

We now understand more clearly the level of methodological detail the reviewer was seeking. In the revised manuscript (from line 581 onward), we expanded the description of the method. In particular, we define the tetrapeptide “weights” and the windows sizes used.

While I appreciate the effort the authors have made to address this point, and I do find

it sufficiently detailed now, I unfortunately do not agree that it is better explained via

1code. While mathematics is a universal language (to a point), Python is not, and it is

not beyond imagining that there are potential readers of the paper who do not know

it. Using code makes the definitions not self-contained, since many of the variables are

not previously defined (presumably being defined somewhere else in their code), and

neither are the functions, so that the reader must either know what they do already

or look them up somewhere else, neither of which is a desirable situation. It also

makes things harder to read since there are also many lines which are unnecessary for

understanding (e.g., “sequence = translate sequence(sequence)” or “from sklearn.svm

import LinearSVC”). In my opinion, self-contained pseudocode is acceptable, but

straight code is not.

Answer:

Our initial intention was to describe these steps using the actual implementation logic, as we felt this would be more accessible to the broad PLOS One readership than formal mathematical notation. However, we acknowledge the reviewer’s concern that including raw Python code within the manuscript is not ideal, as it is not self-contained and may hinder readability.

We therefore fully agree with the reviewer on this point and have replaced the Python code with self-contained pseudo-code, with clearly defined inputs, outputs, and intermediate variables. This presentation removes implementation-specific details while preserving the precise computational logic, and we believe it substantially improves clarity and accessibility for readers from different backgrounds.

While we respectfully maintain that formal mathematical expressions are not strictly necessary for this journal and audience, we believe that the revised pseudo-code, combined with the expanded textual explanation, now provides a sufficiently precise and transparent description of the method.

The full, executable implementation remains available on GitHub for readers who wish to explore or reproduce the analysis in detail.

[Reviewer:]

that I do have an idea what is going on here I can comment a bit on the method

itself. I think it is largely sensible, however some of the names are poorly chosen, which

contributed to my confusion in earlier versions which were not as fully explained. The

“regional decision weight” does not appear to be a “regional” weight at all, but a posi-

tional weight which is then summed over a region to create a regional weight. Likewise,

the “area under the curve” does not appear to be the area under any curve (and cer-

tainly does not correspond to the traditional definition of AUC), but a cumulative sum

of the positional weights. I think both these concepts are valid, but would suggest that

they be renamed.

Answer:

We agree that some of the original names were potentially misleading and may have contributed to earlier confusion. In response, we have renamed the term “area under the curve” with “Cumulative Score” (CS). The term "regional decision weight” was replaced by “positional weight”.

Minor comments:

• l132: space before “[17]”

Done

• l142: “mapping of all of them”, not “mapping all of them”

Done

• l197: Since the method with the lowest runtime is always the linear SVM, I do not see

why the authors cannot just say that they recommend the linear SVM.

Good point, done

• l283: opening double-quotes for “ “Known” ”

Fixed opening and closing double-quotes - “Known”

• l309: “subtelomeric”

Fixed

• l384: space before “(average of 48)”

Added

• l396: Specify the statistical test used here.

Added that those were Welch Two Sample t-tests.

• l398: It is interesting, and perhaps worth mentioning, that although the differences

in Table 6 (bottom) are less significant, there is still a clear separation between the

regions, with East African countries almost all on the bottom and all Asian countries

on the top. I wonder if a rank-sum test could be used here.

As the focus is on the tool, we prefer to keep it as with the focus on the top table.

• l467: “var gene biology”

Split gene & biology

• l470: space before “[4]”

Space added

2• l481–3: “More broadly” is used in two sentences in a row, which sounds awkward.

Changed the second to “More generally”

• l492: capitalise “Python”

We capitalised all occurences

• l546: “var gene sequences”

Fixed to singular

• l583: “two classes, var group B/C or...”

Added the comma

• l592: “mentioned in the Methods”, not “mentioned in the method”

Done

---

## [Decision Letter · Decision Letter 3]

23 Feb 2026

upsML: A high-accuracy machine learning classifier for predicting Plasmodium falciparum var gene upstream groups

PONE-D-25-33506R3

Dear Dr. Thomas D Otto,

We’re pleased to inform you that your manuscript has been judged scientifically suitable for publication and will be formally accepted for publication once it meets all outstanding technical requirements.

Kind regards,

Kokouvi Kassegne

Academic Editor

PLOS One

Additional Editor Comments (optional):

The authors have satisfactorily addressed all the comments. Two minor typos:

Fig 4 still has "AUC"

l599 "extract weights from"

Reviewers' comments:

Reviewer's Responses to Questions

**Comments to the Author**

Reviewer #3: All comments have been addressed

2. Is the manuscript technically sound, and do the data support the conclusions?

Reviewer #3: Yes

3. Has the statistical analysis been performed appropriately and rigorously?

Reviewer #3: Yes

4. Have the authors made all data underlying the findings in their manuscript fully available?

Reviewer #3: Yes

5. Is the manuscript presented in an intelligible fashion and written in standard English?

Reviewer #3: Yes

Reviewer #3: The authors have satisfactorily addressed my comments. Two minor typos:

Fig 4 still has "AUC"

l599 "extract weights from"

**Do you want your identity to be public for this peer review?** For information about this choice, including consent withdrawal, please see our For information about this choice, including consent withdrawal, please see our Privacy Policy .

Reviewer #3: No

---

## [Editor Report · Acceptance letter]

PONE-D-25-33506R3

PLOS One

Dear Dr. Otto,

I'm pleased to inform you that your manuscript has been deemed suitable for publication in PLOS One. Congratulations! Your manuscript is now being handed over to our production team.

Kind regards,

on behalf of

Dr. Kokouvi Kassegne

Academic Editor

PLOS One